# Long-Term Evaluation of Changes in Kidney Function after Switching from Tenofovir Disoproxil Fumarate to Tenofovir Alafenamide in Patients Living with HIV

**DOI:** 10.3390/pharmacy10060164

**Published:** 2022-11-30

**Authors:** Jared M. Gilbert, Kirsten Vest, Troy D. Kish

**Affiliations:** 1James J. Peters Veterans Affairs Medical Center, New York, NY 10468, USA; 2Arnold and Marie Schwartz College of Pharmacy and Health Sciences (LIU Pharmacy), Long Island University Brooklyn, Brooklyn, NY 11201, USA

**Keywords:** HIV, tenofovir, kidney disease, lipids

## Abstract

Tenofovir is one of the most widely used medications for HIV treatment and is administered as either tenofovir disoproxil fumarate (TDF) or tenofovir alafenamide (TAF). Use of TAF is preferred as it is associated with fewer negative impacts on renal function; however, long-term follow-up beyond 96 weeks is limited. A retrospective chart review of patients ≥18 years who received TDF-containing anti-retroviral therapy (ART) for ≥6 months and then switched to a TAF-containing regimen between 1 December 2015 and 1 January 2020 is presented. The primary objective was to evaluate changes in kidney function as measured by eGFR and Scr. The secondary objective was to evaluate changes in lipids. Among the 142 patients identified, the median age was 66 years old with a median follow-up of 3.6 years. The change in kidney function was a median increase in Scr of 0.1 mg/dL and a decrease in eGFR of −8 mL/min/1.73 m^2^. The change in lipid panels at the end of the medication use evaluation endpoint was a decrease in total cholesterol, LDL, HDL, and triglycerides of −2.5, −0.1, −0.6, and −9 mmol/L, respectively. There was no clinically meaningful difference in kidney function as measured by eGFR or Scr, nor was there any clinically meaningful difference in lipid panels in patients switched from TDF to TAF-containing ART. Our observations suggest that the favorable impact of TAF on kidney function is sustained for at least 44 months after conversion from TDF.

## 1. Introduction

Tenofovir is one of the most widely used medications for HIV pre-exposure prophylaxis (PrEP), HIV treatment, and hepatitis B treatment. Tenofovir is an acyclic nucleotide diester analog of adenosine monophosphate, which is administered orally as TDF or tenofovir alafenamide (TAF) [1]. Both are converted intracellularly to the pharmacologically active moiety, tenofovir-diphosphate. Once activated, tenofovir acts with different mechanisms including the inhibition of viral polymerase causing chain termination and the inhibition of viral synthesis. 

TDF can lead to kidney impairment characterized by increases in serum creatinine, proteinuria, glycosuria, hypophosphatemia, and acute tubular necrosis [2]. The main mechanism of TDF-induced nephrotoxicity is through the cellular accumulation of tenofovir from increased entry from the human organic anion transporters (OAT1 and OAT3) and decreased efflux into the tubular lumen. As such, patterns of kidney injury include proximal tubular dysfunction, Fanconi syndrome, acute kidney injury, chronic kidney disease (CKD), and nephrogenic diabetes insipidus. Assessing and monitoring kidney function at baseline and during TDF treatment are the standards in the prevention of TDF-induced nephrotoxicity [3]. Patient risk factors for the development of kidney injury from TDF include elevated serum creatinine, older age, advanced HIV infection, low body weight, hepatitis C coinfection, and prescription of concurrent nephrotoxic drugs. TDF is generally avoided in those with a baseline eGFR <60 mL/min/1.73 m^2^. However, if TDF must be used in this setting, the manufacturer recommends the dose be adjusted when the eGFR falls below 50 mL/min/1.73 m^2^. Adjusting the dose or switching to a less nephrotoxic agent is of clinical importance given that TDF-induced nephrotoxicity was reported in nearly 15 percent of patients treated with TDF for 2–9 years [4]. A study in more than 10,000 HIV-positive patients showed that for each year of exposure to TDF, the risk of proteinuria, rapid decline in kidney function, and developing CKD increased by 34%, 11%, and 33%, respectively [5].

TAF is a prodrug that is absorbed more quickly than TDF and produces higher levels of the active drug, known as tenofovir diphosphate, in cells [6]. Pharmacokinetic studies have shown that administration of 25 mg of TAF, compared with 300 mg of TDF, results in 86% lower plasma tenofovir exposure. This means TAF can be given in smaller doses, leading to lower drug levels in the blood and less exposure for the kidneys, bones, and other organs. Mechanistically, an in vitro study found that in contrast to TDF, TAF is not a substrate of OAT1 or OAT3 and is therefore unlikely to accumulate in kidney proximal tubule cells in an OAT-dependent manner, thereby contributing to the reduced risk of nephrotoxicity [7]. However, comparative data from clinical trials suggest that switching from TDF to TAF is associated with greater increases in total cholesterol, low-density lipoprotein cholesterol, and triglycerides [8]. Nonetheless, current ARV guidelines favor the use of TAF instead of TDF for HIV-infected patients at a high risk for kidney disease [9].

Previous studies have established that switching to TAF was associated with increases in eGFR of 1.5 mL/min if the baseline eGFR was 60–89 mL/min, and 4.1 mL/min if <60 mL/min [10]. In contrast, eGFR decreased by 5.8 mL/min (95% CI, 2.3–9.3) with continued use of TDF in individuals with baseline eGFR <60 mL/min. Further, it was established that baseline eGFR was a significant predictor of the change in eGFR, suggesting that patients on TDF with poorer baseline kidney function would benefit more from switching to TAF [11]. Ultimately, reducing the risk of developing nephrotoxicity is clinically significant given that HIV-1 infection requires life-long treatment. In the chronic management of HIV, pharmacists can play a key role in disease monitoring, management, and risk reduction. Knowledge of agents that are associated with favorable long-term safety profiles can help guide clinicians in appropriate decision-making. 

The primary objective of this medication use evaluation is to evaluate the change in kidney function as measured by eGFR and Scr in patients switched from TDF to TAF-containing ARV regimens. The secondary objective is to evaluate the change in lipids in patients switched from TDF to TAF-containing ARV regimens.

## 2. Materials and Methods

Pharmacists conducted an electronic chart review of patients ≥18 years with HIV who received TDF-containing ART for ≥6 months and then switched to a TAF-containing regimen at any point between 1 December 2015 and 1 January 2020. Patients who did not directly switch from TDF to TAF, had an interruption in ART > 1 month, or did not have a Scr and eGFR value > 9 months after the switch date were excluded. 

Collected demographic information included age, race, ethnicity, sex, chronic kidney disease diagnosis, diabetes mellitus diagnosis, and hypertension diagnosis. ART data collected consisted of the date started on the ART regimen containing TDF, the date stopped the ART regimen containing TDF, the duration of therapy on the ART regimen containing TDF, the date started the ART regimen containing TAF, the date stopped the ART regimen containing TAF (in the event TAF was discontinued), viral load while on TDF within 6 months prior to the switch date (closest value to switch date), viral load > 6 months after switch date, CD4+ count while on TDF within 6 months prior to the switch date (closest value to switch date), and CD4+ count > 6 months after the switch date. Viral load of HIV was determined using COBAS^®^ AmpliPrep/COBAS^®^ TaqMan^®^ HIV-1 Test (Roche, NY, USA). 

Kidney function values collected consisted of an Scr value while on TDF within 6 months prior to the switch date (closest value to switch date), any available Scr values at 6 ± 1 month, 9 ± 1 month, 12 ± 1 month, 18 ± 1 month, 24 ± 1 month, and 36 ± 1 month after the switch date, eGFR while on TDF within 6 months prior to the switch date (closest value to switch date), and any available eGFR at 6 ± 1 month, 9 ± 1 month, 12 ± 1 month, 18 ± 1 month, 24 ± 1 month, and 36 ± 1 month after the switch date. Finally, lipid profiles collected consisted of total cholesterol, LDL, and TG while on TDF within 6 months prior to the switch date (closest value to switch date), and total cholesterol, LDL, and TG > 6 months after the switch date. Descriptive statistics were used to analyze the results. 

## 3. Results

A total of 142 patients were identified who were ≥18 years and received TDF-containing ART for ≥6 months and then switched to a TAF-containing regimen at any point between 1 December 2015 and 1 January 2020. The median age was 66 years old with 63% identifying as Black or African American and 27% identifying as Hispanic or Latino. Of these patients, 48% had hypertension, 22% had diabetes mellitus, and 9% had chronic kidney disease. A complete list of baseline characteristics is found in Table 1.

Of the TDF containing pre-switch regimens, 27% were Atripla (Efavirenz/Emtricitabine/TDF), 19% were Complera (Emtricitabine/Rilpivirine/TDF), 22% were Stribild (Elvitegravir/Cobicistat/Emtricitabine/TDF), 3% were TDF 300mg tablets, and 29% were Truvada (Emtricitabine/TDF). Of the TAF post-switch regimens, 6% were Biktarvy (Bictegravir/Emtricitabine/TAF), 15% were Descovy (Emtricitabine/TAF), 42% were Genvoya (Elvitegravir/Cobicistat/Emtricitabine/TAF), and 36% were Odefsey. 

The average duration of therapy on a TDF-containing regimen was 6.5 ± 3.6 years, and the average duration of therapy on a TAF-containing regimen was 3.8 ± 1.0 years. Baseline kidney function parameters before the switch in therapy showed a median Scr of 1.08 mg/dL and an average eGFR of 84.2 ± 23.2 mL/min/1.73m^2^. The median baseline viral load was <20 copies/mL, and the median CD4+ count was 517 cells/mm^3^. Baseline lipid profiles showed the median total cholesterol was 176 mmol/L, the median LDL cholesterol level was 98.8 mmol/L, the median HDL cholesterol level was 45 mmol/L, and the median triglyceride level was 136 mmol/L. A complete list of baseline characteristics is further summarized in Table 1. 

Baseline median kidney function, lipid panels, HIV viral load, and CD4+ count were compared to the post-switch values. Post-switch values were the last documented values after the 36 ± 1 month interval. The median post-switch time was 44 months. Table 2 and Table 3 confer the same clinically insignificant changes in both kidney function and lipid panels. Median total cholesterol, LDL, HDL, and triglyceride levels all decreased by 2.5 mmol/L, 0.1 mmol/L, 0.6 mmol/L, and 9 mmol/L, respectively, at the median time of 44 months. Viral load and CD4+ count were also assessed, and no appreciable changes were observed.

Median change in kidney function from baseline across the 6 ± 1 month, 9 ± 1 month, 12 ± 1 month, 18 ± 1 month, 24 ± 1 month, 36 ± 1 month, and 44-month intervals was subsequentially grouped by initial TDF regimen as seen in Table 4 below and by transitioned TAF regimen as seen in Table 5 below. Patients transitioned off Atripla (Efavirenz/Emtricitabine/TDF) and Complera (Emtricitabine/Rilpivirine/TDF) had the sharpest decline in kidney function relative to the other TDF regimens with a decrease in eGFR of 11 mL/min/1.73m^2^ and 8 mL/min/1.73m^2^, respectively. Scr also increased in these groups by 0.2 mg/dL and 0.1 mg/dL, respectively, whereas the other TDF-containing ART groups had no change in Scr. Patients transitioned onto Odefsey (Emtricitabine/Rilpivirine/TAF) and Biktarvy (Emtricitabine/Bictegravir/TAF) had the sharpest decline in kidney function relative to the other TAF regimens with a decrease in eGFR of 11 mL/min/1.73m^2^ and 9 mL/min/1.73m^2^, respectively. The sample size was too small to create clusters focusing on one specific TDF-containing ART regiment to one specific TAF-containing ART.

## 4. Discussion

Switching from TDF-containing ART to TAF-containing ART did not appear to result in a clinical impact patient on kidney function or lipid panels over a median of 44 months of observation. There are several potential explanations for these findings. First, aging is a naturally developing biological process associated with gradual structural changes and functional loss of most systems and is often characterized by declining capacity and increasing morbidity [12]. As one of the fastest-aging organs, the kidney shows an age-related reduction in some structures and functions. The annual decrease in kidney parenchyma is about 1%, and the decline of creatinine clearance (CrCl) or glomerular filtration rate (GFR) is approximately 1.0 mL/min per 1.73 m^2^ per year in elderly subjects [13]. Regardless of a patient’s prior ART exposure or their current ART regimen, kidney function would be expected to continue to decline. This is especially a consideration for this medication use evaluation given the extensive median time of 3.7 years that we followed patients for, and given the fact that we followed a much older cohort than in previously published studies. 

These findings have been corroborated in previous studies. One retrospective analysis was performed of a cohort from a publicly funded sexual health clinic in Sydney, Australia [11]. A total of 79 HIV-positive patients were identified that had switched from a TDF- to TAF-containing antiretroviral regimen and were included in the analysis. Almost 90% of patients were male, with a median age of 44 years (IQR 34–53). All pre and post-switch measurements occurred 3 to 12 months apart. Prior to switching from their TDF-containing regimens, patients had an eGFR of 95 ± 2 mL/min/1.73 m^2^. Switching to a TAF-containing regimen did not lead to a significant change in eGFR (mean difference—2.1 mL/min/1.73 m^2^, 95% CI—4.3 to 0.1, *p* = 0.062). While our medication use evaluation corroborates with these findings, it adds to the clinical picture a much older cohort with a median age of 66 years versus this study’s median age of 44 years. Likewise, this study had a majority of patients in the cohort that were on a regimen containing one of dolutegravir, cobicistat, or rilpivirine, all of which have been shown to inhibit creatinine secretion from the proximal kidney tubule, without affecting glomerular filtration rate. Their sample size of 79 did not allow for an exploration of the effect of other agents in the ART regimens. This is of particular importance in real-world clinical settings given there is likely to be more heterogeneity in ART regimens than in clinical trials. Nonetheless, with a slightly larger sample size of 142 patients, our medication use evaluation was able to create clusters based on which TDF or TAF-containing ART patients were on. While we found patients who transitioned off Atripla (Efavirenz/Emtricitabine/TDF) had the sharpest decline in eGFR relative to the other TDF-containing regimens and patients who transitioned onto Descovy (Emtricitabine/TAF) had the sharpest decline in eGFR relative to the other TAF-containing regimens, we cannot conclude these specific agents have a worse kidney profile than the others given that the sample size was too small to track specific clusters taking into account the TDF and TAF-containing regimen as an individual cluster. 

Another single-arm, open-label phase 3 study performed at 70 outpatient centers in the United States, Thailand, the United Kingdom, Australia, Spain, France, the Dominican Republic, Mexico, and the Netherlands enrolled virologically suppressed HIV-1-infected subjects with estimated CrCl 30–69 mL/min in a single-arm, open-label study to switch regimens to Stribild (elvitegravir/cobicistat/emtricitabine/tenofovir alafenamide) [14]. The primary endpoint was the change from baseline in eGFR. In this study, 242 patients with a mean age of 58 years were enrolled, 18% of whom were Black or African American, 39% had hypertension, and 14% had diabetes. Through week 48, no significant change in estimated CrCl was observed. While our medication use evaluation had again a more elderly population, 66 years old versus 58 years old, and a predominately higher Black or African American population, 72% versus 18%, this study is of clinical importance as it had a much higher rate of females, 2% versus 20%. Altogether, these studies add to the large clinical picture that there is no clinically significant change in kidney function across a variety of baseline patient characteristics. 

Treatment with TDF has consistently been associated with a lower increase in lipids compared with other regimens in treatment-naive patients [15]. In randomized controlled trials in ART-naive patients, in switch studies, and in a large study of preexposure prophylaxis, levels of LDL and HDL cholesterol and triglycerides were higher in patients receiving TAF than in patients receiving TDF. Notably, total cholesterol to HDL ratios did not differ between patients receiving TAF and those receiving TDF. Similar findings were also seen in an observational study of Greek patients who switched from TDF to TAF. In this evaluation, 62 patients were switched from TDF/FTC/EVG/COBI to TAF/FTC/EVG/COBI with a median follow-up of 14 months, and it was noted there was an 8.9%, 6.7%, 17.1%, and 2.9% increase in total cholesterol, HDL, LDL, and TG, respectively [16]. Interestingly, although clinically insignificant, our observation found a decrease in median total cholesterol, LDL, and triglycerides post-switch. Reasons for this discrepancy may be unclear as different populations were involved and other factors such as diet and use of cholesterol medications were not well characterized. Further evaluations on the role of TAF on lipid parameters represent a future avenue of research. 

With HIV shifting into chronic disease state management, the control of comorbid conditions is essential for improving patient care. Patients with HIV are at increased risk for both cardiovascular disease and kidney disease, which are both associated with an increased risk of mortality [17,18,19]. Our observations suggest that patients receiving TAF experience a similar decline in their eGFR similar to that associated with normal age-related declines in kidney function. Similarly, we did not detect a negative impact on the lipid panel over the follow-up period. These observations suggest that long-term use of TAF should not negatively impact comorbidities. Providers who are managing patients with HIV can consider transitioning patients from a TDF-containing regimen to a TAF-containing regimen to preserve kidney function.

This medication use evaluation has several limitations. As a retrospective chart review, findings are dependent on the assumption of accurate documentation. First, the patient’s “Active Problem” list was used to collect data regarding patients’ baseline disease states. Of note, CKD is defined based on the presence of either kidney damage or decreased kidney function for three or more months, irrespective of the cause. It is commonly underreported in this “Active Problem” list given the complexity of the diagnosis. While eGFR was collected at baseline and could infer CKD diagnosis, this medication use evaluation did not collect data on albuminuria, proteinuria, or proximal kidney tubular function to further confirm and more accurately report a CKD diagnosis. As such, our cohort may have a higher rate of CKD status, and it has been found that pre-switch eGFR was a significant predictor of the magnitude of eGFR change after the switch in regimens. Secondly, our sample size is small; however, our observations were largely in elderly and Black patients, two populations that are often under-represented in clinical trials. While no definitive conclusions can be made from a group of this size, these observations help provide insight into diverse populations and can hopefully be built upon by future research. An additional limitation to this medication use evaluation is the small representation of women, who made up only 2% of our cohort. Given that low body weight, especially in women, is a risk factor for poor kidney outcomes, it is unclear whether these findings can be extrapolated to female patients. Additionally, we did not collect information on Hepatitis B co-infection, use of statin medications, or changes in weight. Statins and weight changes can directly impact a patient’s lipid panels, which could have influenced our findings. Lastly, the patient’s concomitant medications were not collected. Specifically, mediation classes such as angiotensin-converting enzyme inhibitors (ACE-I), angiotensin receptor antagonists (ARB), and sodium-glucose cotransporter 2 (SGLT2) inhibitors can have a beneficial impact on the progression of kidney disease. Additionally, the use of therapies targeting the lipid profile such as statins, ezetimibe, and fenofibrates could have impacted changes in patient labs over this extended time period.

## 5. Conclusions

Our observations suggest that prolonged use of TAF-containing ART regimens was associated with a decline in eGFR similar to what we would expect to occur naturally from age-related changes alone. These findings support the recommendation of the European society guidelines that patients receiving TDF with an eGFR >60 mL/min should be considered for a switch if their eGFR has declined by 5 mL/year for three or more years and/or their eGFR has decreased >25% from baseline. They also support the National Institutes of Health (NIH) guideline’s recommendation that TDF should be used with caution or avoided in patients with baseline kidney disease as prolonged use of TAF does not appear to be associated with an enhanced decline in eGFR. As such, patients with risk factors for worse kidney outcomes such as advanced HIV disease, longer treatment history, low body weight (especially in women), pre-existing kidney impairment, or concomitant use of pharmacokinetically-enhanced regimens that can increase TDF concentrations should be transitioned to a TAF-containing ART or monitored closely if there is a refusal to switch regimens.

## Figures and Tables

**Table 1 pharmacy-10-00164-t001:** Baseline Patient Characteristics and Laboratory Data.

Patient Characteristics (*n* = 142)	
Age—median, year. (range)	66 (33–90)
Benign hypertension—no. (%)	68 (48)
Diabetes mellitus—no. (%)	31 (22)
Chronic kidney disease—no. (%)	13 (9)
**Race—no. (%)**	
American Indian or Alaska Native	1 (0.7)
Asian	1 (0.7)
Black or African American	89 (63)
Native Hawaiian or other Pacific Islander	1 (0.7)
White	40 (28)
Unknown	4 (3)
Declined to Answer	6 (5)
**Ethnicity—no. (%)**	
Hispanic or Latino	38 (27)
Not Hispanic or Latino	103 (73)
Declined to Answer	1 (0.7)
**ART Characteristics**	
TDF-containing regimen—no. (%)	
Atripla (EFV/FTC/TDF)	39 (27)
Complera (FTC/RPV/TDF)	27 (19)
Stribild (EVG/COBI/FTC/TDF)	31 (22)
TDF 300 mg tab + Background Therapy	4 (3)
Truvada (FTC/TDF) + Background Therapy	41 (29)
TAF-containing regimen—no. (%)	
Biktarvy (BIC/FTC/TAF)	9 (6)
Descovy (FTC/TAF)	22 (15)
Genvoya (EVG/COBI/FTC/TAF)	60 (42)
Odefsey (FTC/RPV/TAF)	51 (36)
Duration of therapy—days (± SD)	
TDF-containing regimen	2389.7 ± 1304.6
TAF-containing regimen	1385.6 ± 371.7
**Baseline Laboratory Data**	
Scr—median, mg/dL (range)	1.081 (0.6–2.3)
eGFR—mL/min/1.73m^2^ (± SD)	84.2 ± 23.2
Viral load—median, copies/mL (range)	<20 (<20–362,000)
Undetectable viral load (%)	102 (71.8)
CD4+ count—median, cells/mm^3^ (range)	517 (38–1366)
Total cholesterol—median, mmol/L (range)	176 (100–334)
LDL—median, mmol/L (range)	98.8 (20.8–220.6)
HDL—median, mmol/L (range)	45 (26.4–139.9)
Triglycerides—median, mmol/L (range)	136 (31–470)

**Table 2 pharmacy-10-00164-t002:** Median renal function, lipid panels, and HIV viral/CD4+ count pre and post-switch from TDF to TAF containing ART.

Lab	Pre-Switch Value— Median (Range)	Post- Switch Value *—Median (Range)
eGFR (mL/min/1.73m^2^)	84 (29–154)	79 (37–138)
Scr (mg/dL)	1 (0.6–2.3)	1.1 (0.6–1.9)
Total cholesterol (mmol/L)	176 (100–334)	175 (72–263)
LDL (mmol/L)	98.8 (20.8–220.6)	93.5 (22.8–174.4)
HDL (mmol/L)	45 (26.4–139.9)	43 (24–99)
Triglycerides (mmol/L)	136 (31–470)	115 (54–429)
Viral load (copies/mL)	<20 (<20–362,000)	<20 (<20–1,230,000)
CD4+ count (cells/mm^3^)	517 (38–1366)	532 (42–1285)

* Median time of 3.7 years (0.7–5.2).

**Table 3 pharmacy-10-00164-t003:** Median change in viral load, CD4+ count, and lipid panels.

Lab	Median Change (Range)
Viral load (copies/mL)	<20 (−361,747–1,230,000)
CD4+ count (cells/mm^3^)	4 (−786–540)
Total cholesterol (mmol/L)	−2.5 (−113–85)
LDL (mmol/L)	−0.1 (−96.6–86.5)
HDL (mmol/L)	−0.6 (−40.9–49)
Triglycerides (mmol/L)	−9 (−213–218)

**Table 4 pharmacy-10-00164-t004:** Median change (range) in Scr (mg/dL) and EGFR (mL/min/1.73m^2^) from baseline grouped by initial TDF regimen.

Time (mo)	Atripla (*n* = 39)		Complera (*n* = 27)		Stribild (*n* = 31)		TDF + Background Therapy (*n* = 4)		Truvada + Background Therapy (*n* = 41)	
	∆ Scr	∆ eGFR	∆ Scr	∆ eGFR	∆ Scr	∆ eGFR	∆ Scr	∆ eGFR	∆ Scr	∆ eGFR
**6 ± 1**	0 (−0.4–0.3)	0 (−47–23)	0 (−0.5–0.1)	0 (−13–36)	0 (−0.3–0.3)	0 (−37–35)	0 (−0.1–0.3)	0 (−14–6)	−0.05 (−0.5–0.2)	4 (−28–46)
**9 ± 1**	0 (−0.5–0.5)	0 (−67–27)	0 (−0.5–0.3)	0 (−24–45)	0 (−0.5–0.1)	0 (−12–23)	−0.1 (−0.3–0.1)	8.5 (−6–23)	0 (−0.7–0.7)	0 (−36–41)
**12 ± 1**	0.1 (−0.7–0.5)	−10 (−67–35)	0.1 (−0.3–0.5)	−4 (−51–36)	0 (−0.4–0.4)	0 (−29–20)	0 (−0.2–0.6)	0 (−23–14)	0 (−0.5–0.4)	0 (−37–20)
**18 ± 1**	0.1 (−0.7–0.5)	−9 (−59–35)	0 (−0.3–0.3)	0 (−30–36)	0 (−0.4–0.3)	−1 (−16–17)	0.1 (0–0.2)	−10 (−17–0)	−0.05 (−1.3–0.2)	4 (−23–26)
**24 ± 1**	0.1 (−0.6–0.6)	−14 (−59–27)	0 (−0.3–0.3)	−1 (−30–30)	0 (−0.5–0.3)	−3.5 (−24–20)	0 (−0.1–0.3)	−0.5 (−14–6)	0 (−0.7–0.5)	−1 (−38–56)
**36 ± 1**	0.1 (−0.5–0.5)	−10 (−60–20)	0.1 (−0.2–0.3)	−8 (−20–17)	0 (−0.2–0.4)	−1 (−23–20)	0.1 (0.1–0.1)	−6 (−6–−6)	0 (−0.7–0.4)	−1 (−41–20)
**44**	0.2 (−0.6–0.5)	−11 (−60–26)	0.1 (−0.4–0.7)	−8 (−37–54)	0 (−0.5–0.3)	−1 (−23–23)	0.15 (0–0.2)	−11 (−17–−1)	0 (−0.5–0.6)	−1 (−47–45)

**Table 5 pharmacy-10-00164-t005:** Median change (range) in Scr (mg/dL) and EGFR (mL/min/1.73m^2^) from baseline grouped by transitioned TAF regimen.

Time (mo)	Biktarvy (*n* = 9)		Descovy (*n* = 22)		Genvoya (*n* = 60)		Odefsey (*n* = 51)	
	∆ Scr	∆ eGFR	∆ Scr	∆ eGFR	∆ Scr	∆ eGFR	∆ Scr	∆ eGFR
**6 ± 1**	0.1 (−0.1–0.3)	−9.5 (−45–15)	−0.1 (−0.3–0.2)	8.5 (−18–46)	−0.05 (−0.5–0.3)	3 (−47–35)	0 (−0.5–0.2)	0 (−28–36)
**9 ± 1**	0 (−0.1–0.3)	0 (−45–11)	0.1 (−0.1–0.3)	−10 (−24 −8)	0 (−0.7–0.7)	0 (−67–45)	0 (−0.5 –0.3)	0 (−17–34)
**12 ± 1**	0.1 (−0.1–0.4)	−9 (−29–9)	0 (−0.3–0.4)	0 (−37–21)	0 (−0.5–0.6)	0 (−67–20)	0 (−0.2–0.6)	0 (−23–14)
**18 ± 1**	0 (0–0.3)	−1 (−45–0)	−0.05 (−1.3–0.2)	5.5 (−18–26)	0 (−0.5–0.5)	0 (−59–20)	0 (−0.7–0.5)	−1 (−51–36)
**24 ± 1**	0.1 (0–0.1)	−4.5 (−11–20)	0.1 (−0.3–0.5)	−10 (−31–46)	0 (−0.7–0.4)	−1 (−59–56)	0.1 (−0.6–0.6)	−7 (−38–30)
**36 ± 1**	0.2 (−0.1–0.3)	−20 (−20–11)	0 (−0.2–0.4)	−1 (−18–11)	0 (−0.7–0.5)	−1 (−60–20)	0.1 (−0.5–0.3)	−9.5 (−37–20)
**44**	0.1 (−0.2–0.3)	−9 (−46–20)	0 (−0.4–0.3)	−1 (−31–45)	0.1 (−0.5–0.6)	−3.5 (−60–40)	0.1 (−0.6–0.7)	−11 (−37–54)

## Data Availability

The data presented in this study are available on request from the corresponding author. The data are not publicly available due to institutional policy.

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
