# Peer review of "Long-Term Evaluation of Changes in Kidney Function after Switching from Tenofovir Disoproxil Fumarate to Tenofovir Alafenamide in Patients Living with HIV"

_pharmacy, 2022, doi:10.3390/pharmacy10060164_

Round 1
Reviewer 1 Report
The current study is quite interesting to read and addresses a very relevant topic, which is how slight medication changes can affect kidney function, even if concerning the same drug molecule. It is also a long-term study (5 year period), which makes it quite robust. Nevertheless, there are a few changes that should be made before acceptance for publication:
- Although 142 patients is a reasonably sized sample, you should address the limitations in what concerns drawing generalized conclusions;
- In the abstract there should be a “final conclusion”, which should be written at the end;
- Were the patients from a specific demographic origin (country)? If yes, then this should be stated, and it should also be mentioned as a limitation of the study;
- The title of the tables should be more descriptive of their content (specially Table 1);
- More references should be used to support the introduction and discussion sections.
Author Response
1) Added to limitations: "Secondly, our sample size is small; however, our observations were largely in elderly and Black patients, two populations that are often under-represented in clinical trials. While no definitive conclusions can be made from a group of this size, these observations help provide insight into diverse populations and can hopefully be built upon by future research. "
2) Added a conclusion statement to the abstract " Our observations suggest that the favorable impact of TAF on kidney function is sustained for at least 44 months after conversion from TDF."
3) All patients were from a US Veteran's Affairs hospital. Country of origin was not collected but Race and Ethnicity data is reported in Table 1. A majority of patients were Black in our analysis.
4) Changed title to Baseline Patient Characteristics and Laboratory Data. Also added sub-titles within the table to break up the sections
5) Included new reference and data regarding impact on lipid profile post conversion to TAF.
Reviewer 2 Report
Did the patients present co-infection with hepatitis B virus?
What were the reasons for switching from TDF to TAF?
Pacientii in varsta prezinta cel mai adesea dislipidemie. Pacientii din studiu utilizau medicamente hipolipemiante?
Even if this study has limitations, it brings valuable information regarding the possibility of changing TDF with TAF in elderly patients
Author Response
1) We did not collect concomitant Hepatitis B in the patient's medical history. Added a statement in the limitations paragraph.
2) Reasons for switching from TDF to TAF were not collected as part of this review. As all patients were stable on the TDF combination, we assume the switches were made for the favorable renal and bone health profile of TAF when compared to TDF.
Reviewer 3 Report
As a physician who regularly cares for patients with HIV infection, I understand the importance of using medication with few long-term adverse effects, since the medication is chronic.
The authors have compared the renal function of patients taking regimens with TDF, and have compared it after switching to TAF, not finding statistically significant differences.
In the abstract (lines 12-20) the results of the changes are presented, objectifying a worsening of renal function and an improvement of the lipid profile. In the subsequent lines it is explained that the changes were not statistically significant. However, the wording can be misleading. It would be convenient to add the result of the statistical analysis to the results directly.
In material and methods (page 2 lines 84-88) the objectives, already exposed in the introduction (lines 75-78), are repeated. They should be removed to avoid duplication, or state more detailed objectives that justify repetition.
The statistical methods used have not been specified. A paragraph on this should be included.
Table 1 shows a viral load of 0 copies/ml. Viral load is usually considered undetectable with <20 or <50 copies/mL, depending on the technique used. The technique used must be specified in material and methods.
Median viral load was 0 copies/mL, however the range is wider. The number and percentage of patients with undetectable viral load should be specified, being more informative of the characteristics of the patients than the median viral load.
Table 1 also quantifies the patients with TDF/FTC and TAF/FTC, but the accompanying regimen is not specified, and dual therapy with this medication is not indicated for treating the infection. Was the use of pre-exposure prophylaxis medication considered?
Almost half of the patients were hypertensive, and almost a quarter of them were diabetic. Both the influence of these pathologies and the specific medication have not been evaluated. Do the authors consider that it can significantly influence the results obtained?
63% of patients were black or African-American, with a corresponding influence of race on SCr measurements. Has it been related pathology, such as HIV nephropathy, considered by the authors?
To evaluate the lipid profile, total cholesterol, LDL, and triglycerides were considered, without considering the use of statins and other antilipid-lowering drugs. Weight gain has been described in changes to TAF/FTC/BIC. Has this been studied by the authors?
In tables 2 and 3, and later in tables 4 and 5, the statistical results are not specified. If the study is only descriptive observational, it should be specified in material and methods.
The conclusions should explain why, if there are no clinically significant changes, the results support the guideline recommendations to switch from TDF to TAF.
Author Response
1) Statistical analysis was not conducted on these parameters as there was no comparator group. The abstract stated there was no "clinically significant" changes in SCr or eGFR. I have changed this wording to read "clinically meaningful" so as to not mislead the reader.
2) Deleted duplication of objectives.
3) Conclusions were derived based on descriptive statistics as there was no comparator TDF group for which to compare outcomes. Added a statement to the methods section stating this.
4) Changed value of 0 to < 20 as per our laboratory's measurements and added statement " Viral load of HIV was determined using COBAS® AmpliPrep/COBAS® TaqMan® HIV-1 Test (Roche). " Additionally, added a row in the Baseline Characteristics table that shows the percentage of patients who were undetectable at baseline. With the focus of this assessment being the evaluation of the safety of TAF and not the efficacy, we did not evaluate HIV control over the period of time. The efficacy of TAF has been already demonstrated in larger publications.
5) Patients in the TDF 300 mg and FTC/TDF groups were all receiving additional antiviral medications as per their prescriber's choice. All patients in this observation had a diagnosis of HIV so no PrEP patients were included. Added the clarifier to the tables to state that patients were receiving that form of TDF + Background Therapy
6) Control of hypertension and diabetes as well as use of select medication in these disease states (ACE-I, ARB, SGLT2, etc) can influence kidney disease progression. We did not evaluate these parameters given the challenges associated with assessing nearly 4 years of disease control and that there would be incomplete information on this based on how frequently patient's followed-up with their providers. Additionally, these confounders were not addressed in the phase 3 clinical trials used to evaluate TAF's impact on the kidneys (Sax PE, et al. Lancet;2015;385:2606).
7) Both black race and HIVAN could be contributors to kidney function interpretation. Unfortunately, with the retrospective nature, we were unable to evaluate diagnosis of HIVAN or presence of proteinuria.
8) The reviewer is correct in that we did not evaluate the concomitant use of statins or changes in weight, both of which can influence lipid panel readings. Added an acknowledgement of this in the limitations section.. " Additionally, we did not collect information Hepatitis B co-infection, use of statin medications, or changes in weight. Statins and weight changes can directly impact patient’s lipid panels which could have influenced our findings. "
9) Descriptive statistics were only used for this evaluation as there was not a comparator group who remained on TDF. A statement of this was added to the methods section.
10) Added a statement in conclusion that our findings support guideline recommendations to convert patients to TAF because our observations suggest that TAF is not associated with more faster progression of CKD (elevation in SCr or decline in eGFR).